# Step-by-Step Replacement of Cyano Groups by Tricyanovinyls—The Influence on the Acidity

**DOI:** 10.3390/molecules28248157

**Published:** 2023-12-18

**Authors:** Agnes Kütt

**Affiliations:** Institute of Chemistry, University of Tartu, Ravila 14A, 50411 Tartu, Estonia; agnes.kutt@ut.ee

**Keywords:** cyanocarbon acids, tricyanovinyl, polycyano compounds, gas-phase acidity, COSMO-RS, acid-base properties

## Abstract

Acid-base properties are the simplest expression of compounds’ coordinating ability. In the present work, we studied in silico how the gas-phase Brønsted acidity (GA) of several polycyano-substituted compounds change when cyano (CN) groups are replaced by 1,2,2-tricyanovinyl (TCNV) groups in (iso)cyanic acid, dicyanoamine, cyanoform, and hydrogen tetracyanoborate. Different tautomers and conformers/isomers are included in this study. Gas-phase acidity values are compared with the acidities of various acids, including percyanated protonated monocarba-*closo*-dodecaborate (carborane acid) and dodecaborate, as well as hydrogen cyanide and 1,2,2-tricyanoethene. An estimation of acetonitrile (MeCN), dimethylsufoxide (DMSO), and 1,2-dichloroethane (DCE) acidities is presented using the COSMO-RS method and correlation analysis. The strongest acid with four TCNV groups shows remarkable acidic properties.

## 1. Introduction

The gas-phase acidity (GA) of a neutral acid AH and proton affinity (PA) of the corresponding anionic base A^–^ refers to the following equation:(1)$$AH\overset{\begin{array}{l} PA\\ GA \end{array}}{⇄}A^{-}+H^{+}$$

PA of an anionic base A^−^ is defined as the enthalpy change (Δ*H*_acid_) for the gas-phase reaction 1 between a proton and the anion to produce the neutral conjugate acid of the anion. GA of an acid HA is the Gibbs free energy change (Δ*G*_acid_) on deprotonation of the acid according to Equation (1). GA values include the entropy factor. GA is directly measurable and comparable with experimental values, and therefore they will be used in discussions in this work. Reaction 1 shows that the smaller the PA or GA value, the more easily the proton is detached from the acid AH; therefore, the stronger the acid is. Acids with GA values around 400–350 kcal/mol can be considered weak acids, acids having GA values lower than 300 kcal/mol are superacids, while acids with a GA value lower than 250 kcal/mol are referred to as hyperacids.

Acid dissociation constant *K*_a_ (usually expressed as its negative logarithm p*K*_a_) is an equilibrium constant that shows how strong an acid AH is according to Equations (2) and (3). S refers to the solvent $,\;\;A^{-}$ refers to an acid anion, and *a* to an activity. The stronger the acid, the lower the p*K*_a_ value. The p*K*_a_ value depends strongly on the solvent [1,2]. According to the definition, a superacid is an acid that is stronger than sulfuric acid in the respective solvent/medium. p*K*_a_ values of superacids are lower than 9 in MeCN, 42 in DCE [3], and around 1.4 p*K*_a_ units in DMSO [4].
(2)$$AH+S\overset{K_{a}}{⇄}A^{-}+SH^{+}$$
(3)$$pK_{a}=-\log\frac{a\left({\mathrm{SH}}^{+}\right)\cdot a\left(A^{-}\right)}{a\left(\mathrm{HA}\right)}$$

Cyanocarbon chemistry gained its first insights quite some time ago with the seminal works by Webster, Middleton, and Engelhardt introducing different polycyanated compounds such as polycyano cyclopentadienides [5], pentacyanopropenide [6], and other compounds with tricyanovinyl substituents [7]. Already back then, the authors realized that conjugate acids of anionic cyanocarbon compounds are very strong acids.

Although cyanocarbon acids are not applicable as pure acids, the anions still find plenty of recognition. Computational acidity studies of cyanocarbons are not rare, and have been conducted mainly by Vianello and Maksić [8,9,10,11,12], Leito and Koppel [13,14], and others [15]. The high acidifying effect of the cyano (CN) group has been used to probe the boundaries of the conventional acidity of superacids. It has been shown that the acidity of different hydrocarbons can be increased by more than 100 kcal/mol by substituting them extensively with CN groups [8]. The strongest organic cyanocarbon acids reach the gas-phase acidity of 230 kcal/mol.

Substitution with multiple cyano groups has such an acidifying effect because the anion formed after deprotonation is highly stabilized by efficient charge delocalization. The negative charge is attracted by the CN groups both via strong inductive and mesomeric (resonance) effects. Moreover, compared to -NO_2_ or -SO_2_CF_3_ groups, the CN group is very compact, i.e., it has low steric demand. Because the CN group is stick-like, it cannot be “twisted out of plane” of an aromatic system, which is a typical cause of weakening of the acidifying effect of the nitro group, for example. Thus, it is possible to put numerous CN groups into one molecule without creating extensive steric repulsion. For example, pentacyanophenol (p*K*_a_(MeCN) = 4.2) is one of the strongest acids among phenols, leaving 2,4,6-(SO_2_CF_3_)_3_-phenol behind by 1 p*K*_a_ unit and 2,4,6-trinitrophenol by 7 p*K*_a_ units in acetonitrile (MeCN). Inserting five-SO_2_CF_3_ or nitro groups into the benzene moiety proves impossible.

However, the CN group also has some disadvantages when it comes to creating strong acids. From the acidity and coordination perspective, the lone pair of nitrogen attracts all nucleophiles, including the proton. It has been shown computationally that the monocarba-*closo*-dodecaborane acid (CB_11_H_12_H) becomes weaker when one of the hydrogens is substituted with a CN group [16]. However, replacing all 12 positions in the carborane moiety with the CN groups leads to an enormous acidity increase by 50 kcal/mol. Pentacyanocyclopentadienide C_5_(CN)_5_^−^ has been a notorious anion that cannot be protonated even with perchloric acid [5]. Later, it was shown by C. A. Reed that pentacyanocyclopendadiene HC_5_(CN)_5_ has a polymeric structure where anions are bridged with protons [17]. The black-brown solid that was described to form is common when trying to prepare the pure acid from polycyano anions.

The 1,2,2-tricyanovinyl group (TCNV) accommodates three CN groups. From the point of view of the original acidity center, all CN groups are further away; therefore, inductively, they have a lower influence on the acidity. However, the resonance possibility creates additional acidity centers with the adjacent CN groups. The negative charge of the anion has the possibility to further delocalize, stabilizing the anionic and giving the possibility to create stronger acid compared to the system with only CN groups. 

Isocyanic acid **1H**, dicyanamine **3H**, cyanoform **6H** [18], and tetracyanoborate **10** are all well-known compounds and have been chosen as the base structures of current work. Tetracyanoborate **10** is a weakly coordinating anion [19,20], a component for ionic liquids [21,22,23], and its acid form is stable in the diluted solution [3]. The compounds substituted by TCNV groups could be prepared from the readily available tetracyanoethylene (TCNE). Compound **2** can be prepared from TCNE by hydrolysis [6], **4** with ammonium acetate [24], and **5** in liquid ammonia [6]. Compound **7** has been prepared by combining TCNE and malononitrile [6]. Synthesis methods for **8** are available by Webster [7]. Compound **9** is not yet prepared, but the synthesis could be envisaged according to published procedures [25]. Borates with TCNV groups **11**–**14** are not yet prepared.

### Compounds under Study

Four base structures—(iso)cyanic acid **1H**, dicyanamine **3H**, cyanoform **6H**, and hydrogen tetracyanoborate **10H**—are denoted as *initial compounds* (either in acid or anionic form). The CN groups of these compounds were stepwise replaced by TCNV groups (*steps 1 to 4*), as shown in Table 1.

In each case, different tautomers were included in the calculations (Figure 1). If compounds bearing only CN groups are considered, tautomers **T1** and **T2** exist. **T2**, **T4**, and **T5** are imino taoutomers. Tautomers **T1**–**T5** have resonance structures written in neutral form, and **T6** can be written in zwitterionic form. In the case of central atom B, the tautomers do not correspond to the structural formula in Figure 1 due to the peculiarity of boron bonding; they correspond more to zwitterionic resonance structures rather than neutral forms.

Altogether, 14 *title compounds* were studied. Additionally, to compare the acidities, five related compounds were added to the study (Table 2). We look at each anion or acid from the perspective of the *central atom* O, N, C, and B, sometimes in the text also referred to as X. CN and TCNV groups are always looked at as substituents, even if protonated, without further specification of their names. 

In the case of each tautomer, different conformers and isomers (diastereomers, Figure 2) were included in the calculations. The TCNV group is always planar in the case of anions and tautomers **T1**, **T2**, and **T4**–**T5**. For anion and tautomers **T1**, **T2**, and **T6**, the free rotation about the C-X bond can be envisaged. If the torsion angle between C-C-X-C is 0° (marked with red color in Figure 2), then the TCNV group is in *syn* conformation; when the torsion angle is 180°, the *anti* conformer exists. In the case of tautomers **T3**, **T4,** and **T5**, there is a formal double bond between C and X atoms; free rotation is impossible, and therefore *cis* and *trans* isomers are distinguished. In the case of tautomer **T3**, where the proton is attached to the terminal carbon atom, the carbon atom has sp^3^ character, and the TCNV group is not planar. In there, the conformers can be distinguished as *syn*(H) (torsion angle between H-C-C-X atoms is 0°)*, anti*(H) (180°), and *eclipsed*(H) (≈90°), as shown in Figure 2. This type of distinction between different diastereomers has also been used when X = B. Distinguishing between diastereomers and tautomers is only needed for computations to see which form is more or less stable. In reality, a mixture of tautomers and different conformers may exist. This fact is taken into account when solution p*K*_a_ values are calculated.

## 2. Results and Discussion

### 2.1. Gas-Phase Acidities

The final results and the experimental or theoretical literature data are presented in Table 3. The whole set of calculations with the *E*, *H*, and *G* values and COSMO-RS p*K*_a_ values (see below) are available in Appendix A and the data repository [26].

According to the workflow, the DFT BP86/def-TZVP method was used to carry out gas-phase calculations (i.e., geometry optimization, and frequency calculations) for all the tautomers and diastereomers. Altogether, approximately 400 gas-phase calculations were carried out, and more than 300 GA values were obtained (Figure 3). GA values were always calculated for the structurally most similar acid-base pairs. This means, for example, for the *syn-syn* configuration acid, the *syn-syn* structured anion is used. Current GA values do not contain information from different tautomers or conformers; they have been calculated only using the most stable structurally similar forms. 

After the most stable forms were received, other calculation methods were employed to evaluate the suitability of the def-TZVP method (Table 4). These different methods were used only to calculate GA values for the most stable acid-base pairs resulting from the BP86/def-TZVP method. It was decided that GA values obtained from the B3LYP/6-311 + G(d,p) method were to be used as *selected GA values* (Table 3 and Figure 4) to make the results comparable with Maksic, Vianello, and Koppel’s work (e.g., [9,27]). This calculation method has a sufficiently high precision to obtain data for larger molecules and is generally accepted as a precise method for *H* and *G* values [28]. The G4(MP2) method [29], known as the method obtaining exact energies and which matches generally very well with the experimental data, was also tested among other accurate methods, such as W1RO and CBS-APNO; however, all these methods are highly resource-consuming and it was not possible to get results for all the compounds.

**Table 3 molecules-28-08157-t003:** Theoretical (DFT B3LYP/6-311 + G(d,p)) PA and GA values (kcal/mol). All literature values are shown in brackets, whereas values in parentheses have a computational origin. Corrected COSMO-RS p*K*_a_(MeCN) values, p*K*_a_(DMSO), and p*K*_a_(DCE) values are derived from correlation analysis.

No.	Compd		PA [kcal/mol]	GA [kcal/mol]	p*K*_a_(MeCN)	p*K*a(DMSO)	p*K*a(DCE)
**1H**	[CN-O]H	T2	341.0 [341.2 *^a^*]	334.6 [334.7 *^a^*]	24.2	12.4	60.6
**2H**	[O-TCNV]H	T1	289.4	282.1	5.1 [4.39 *^e^* (3.1) *^f^*]	−7.0	38.6 [36.3 *^i^*]
**3H**	[(CN)_2_N]H	T2	311.7	304.5	12.5	0.5	47.1
**4H**	[CN-N-TCNV]H	T2	290.4	283.5	5.7	−6.3	39.3
**5H**	[N-TCNV_2_]H	T1	276.4 [(278.0) *^b^*]	269.1	1.2 [(0.3) *^f^*]	−11.0	34.1 [33.0 *^i^*]
**6H**	[(CN)_3_C]H	T2	293.0 [(293.7) *^b^*]	286.0 [294.8 *^c^*]	4.9 [5.0 *^f^* (5.1) *^f^*]	−7.2	38.4 [38.6 *^i^*]
**7H**	[(CN)_2_-C-TCNV]H	T2	274.4 [(274.8) *^b^*]	266.6	−2.4 [(−2.8) *^f^*]	−14.7	29.9 [29.7 *^i^*]
**8H**	[CN-C-TCNV_2_]H	T2	265.7 [(266.0) *^b^*]	258.6	−5.2	−17.5	26.7
**9H**	[C-TCNV_3_]H	T4	257.7	252.3	−7.6	−20.0	23.9
**10H**	[(CN)_4_B]H	T2	265.6	258.7	−3.5 [(−1.0) *^f^*]	−15.8	28.7 [31.7 *^i^*]
**11H**	[(CN)_3_-B-TCNV]H	T2	260.7	253.9	−4.1	−16.4	28.0
**12H**	[(CN)_2_-B-TCNV_2_]H	T2	256.5	249.3	−4.9	−17.2	27.0
**13H**	[CN-B-TCNV_3_]H	T2	253.3	246.0	−5.9	−18.2	25.9
**14H**	[B-TCNV_4_]H	T6	236.2	229.3	−12.9	−25.3	17.9
**15H**	HCN		349.3 [350.5 *^a^*]	341.9 [343.2 *^a^*]	28.4 [(23.40) *^g^*]	16.8 [12.9 *^h^*]	65.4
**16H**	C(CN)_2_=CH(CN)		330.7	323.2	29.7	18.1	66.9
**17H**	[B_12_(CN)_12_]H^-^		295.1	288.7	−8.1	−20.4	23.4
**18H**	[B_12_(CN)_12_]H_2_		243.9	237.2	−11.4	−23.8	19.6
**19H**	[CB_11_(CN)_12_]H		231.6	224.9 [(225.0) *^d^*]	−14.2	−26.7	16.3

*^a^* Ref. [30] *^b^* Computational values from the ref. [9]. *^c^* Ref. [31]. *^d^* Ref. [32]. *^e^* Ref. [33]. *^f^* p*K*_a_(MeCN) values in parentheses are not directly measured in MeCN, but in DCE, the values are derived from the correlation between the p*K*_a_ of DCE and MeCN [3]. *^g^* Ref. [34]. *^h^* Ref. [35]. *^i^* Ref. [36].

**Figure 3 molecules-28-08157-f003:**
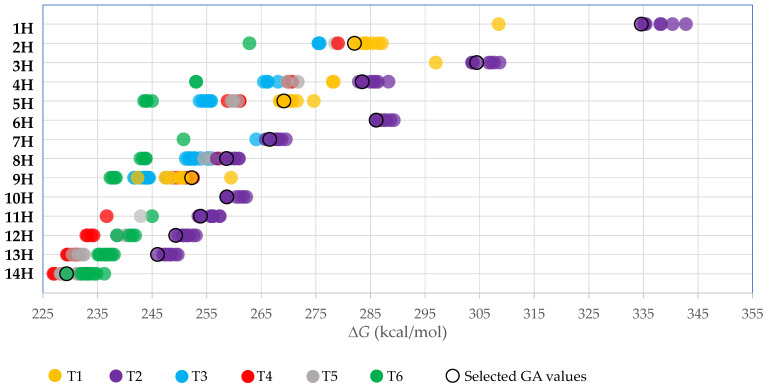
Each circle indicates calculated GA values with all possible color-coded calculation methods based on different tautomers.

**Figure 4 molecules-28-08157-f004:**
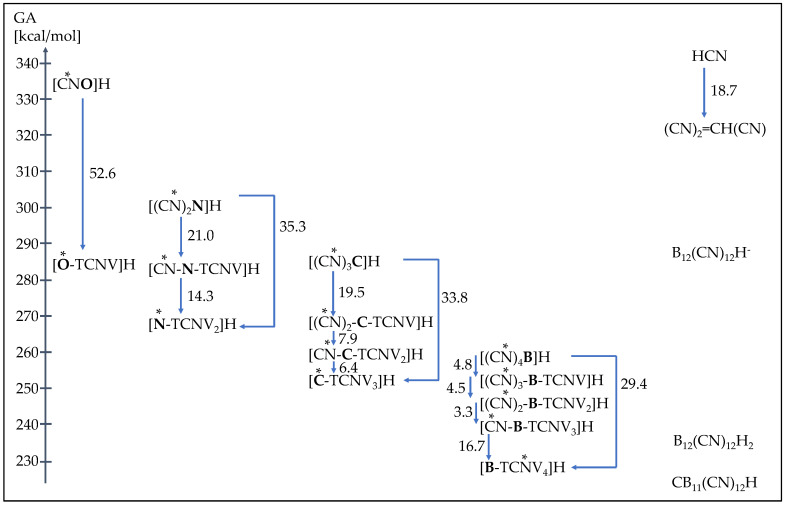
Overview of the change in GA values (B3LYP/6-311 + G(d,p)) in stepwise replacement of CN groups by TCNV groups of title compounds; additional acids are included. Protonation sites are indicated with *.

**Table 4 molecules-28-08157-t004:** GA values obtained using different calculation methods. Assigned GA values are in bold.

Compound		Turbomole				Gaussian			
		BP86/ def-TZVP	BP86/ def2-TZVPP	BP86/ def2-TZVPPD	B3LYP /def2-TZVPPD	B3LYP/ 6-311 + G(d,p)	G4(MP2)	W1RO	CBS-APNO
[CNO]H	**1H**	338.3	340.3	338.1	342.8	**334.6**	335.2	335.1	335.4
[O-TCNV]H	**2H**	282.5	284.2	283.5	284.2	**282.1**	287.1	286.5	285.3
[(CN)_2_N]H	**3H**	307.1	308.6	307.7	306.7	**304.5**	303.8	303.6	303.7
[CN-N-TCNV]H	**4H**	285.0	286.4	285.9	285.5	**283.5**	283.9	283.5	282.9
[N-TCNV_2_]H	**5H**	269.8	270.1	270.3	270.7	**269.1**	274.6		271.5
[(CN)_3_C]H	**6H**	287.9	289.3	288.7	287.5	**286.0**	286.7	286.1	286.0
[(CN)_2_-C-TCNV]H	**7H**	268.1	269.5	268.7	268.0	**266.6**	267.8		265.9
[CN-C-TCNV_2_]H	**8H**	259.6	260.9	259.4	260.9	**258.6**	260.2		256.9
[C-TCNV_3_]H	**9H**	250.6	250.4	248.4	250.1	**252.3**	259.5		
[(CN)_4_B]H	**10H**	261.1	262.2	261.7	260.4	**258.7**	258.8	258.7	258.9
[(CN)_3_-B-TCNV]H	**11H**	256.2	257.4	257.4	255.7	**253.9**	254.1		253.5
[(CN)_2_-sB-TCNV_2_]H	**12H**	251.9	253.0	252.6	251.1	**249.3**	249.8		
[CN-B-TCNV_3_]H	**13H**	248.5	249.7	249.3	248.3	**246.0**	247.0		
[B-TCNV_4_]H	**14H**	233.1	234.4	232.3		**229.3**			
HCN	**15H**	341.9	340.2			**341.9**			
C(CN)_2_=CH(CN)	**16H**	323.6	323.8			**323.2**			
[B_12_(CN)_12_]H^-^	**17H**	291.5	292.3			**288.7**			
[B_12_(CN)_12_]H_2_	**18H**	239.8	241.3			**237.2**			
[CB_11_(CN)_12_]H	**19H**	228.0	229.1			**224.9**			

All the GA values from all the tested calculation methods stay within 11 kcal/mol (Table 4). Usually, they remain between 3 and 5 kcal/mol. The B3LYP/6-311 + G(d,p) calculations result in the highest acidities (i.e., the lowest GA values); however, in most cases, they match pretty well with G4(MP2) GA values. The most significant differences are for the molecules that are extensively substituted with TCNV groups, where perhaps the electron diffusion and its precision in the calculation may play a crucial role. BP86/def-TVZP values, in fact, describe all the title compounds sufficiently well. 

In Figure 3, all the GA values calculated in this work are depicted as circles, color-coded based on different tautomers. It gives an excellent overview of how, moving down to compounds with more TCNV groups, the GA values of different tautomers come closer to each other. In the case of the first five compounds, groups of tautomers lay relatively far from each other. In the case of C-centred molecules, the different GA values of different tautomers are closer or even mixed. For acids **6H**, **7H**, and **8H**, tautomer **T2** is the most stable form. For the acid **9H**, the tautomers **T1**, **T4**, and **T5** have very similar GA values, while the GA values for tautomer **T1** vary widely, ranging from 242 to 251 kcal/mol, depending on the conformation. In all cases, the **T6** tautomer is one of the least stable acids except for hydrogen borates: the **T6** tautomer will become more and more stable compared to **T4** and **T5** tautomers, and when a single CN group is missing, the **T6** tautomer is the most stable acid form. 

In general, when the anion contains a single CN group (**T2**), then it is the most basic site of the anion. When the single CN group is missing, then in three cases out of four (central atom O, N, and C), the **T1** tautomer is the most stable neutral form. In the case of hydrogen borates, **T1** cannot be the most stable form; instead, tautomer **T6** is the most stable acid form. 

Figure 4 presents the overview of the change in GA values during the replacement of CN groups by TCNV groups of initial compounds (B3LYP/6-311 + G(d,p) basis set). It is seen how every additional replacement increases acidity less than the first replacement. The only exception is hydrogen borates, where the initial three replacements increase acidity a humble 3 to 5 kcal/mol, but the last replacement increase it almost to 17 kcal/mol. This phenomenon comes from the fact that the protonation site changes from a relatively stable **T2** tautomer to **T6**. 

Divalent oxygen substituted with the CN group exhibits two tautomers: OH-acid (cyanic acid, **1H-T1**) and 29.8 kcal/mol more stable NH-acid (isocyanic acid **1H-T2**). Replacement of the CN group with the TCNV group in isocyanic acid **1H** leads to an enormous acidity increase by 52.6 kcal/mol, reaching the GA value of 282.1 kcal/mol for **2H-T1**. Such an impressive acidity increase comes from the fact that the size of the anion as well as the charge delocalization possibility increases considerably. Small [CNO]^−^ anion has a limited ability to delocalize the negative charge, and therefore, the acid form does not deprotonate easily. When the TCNV group replaces the CN group, the negative charge is stabilized by extensive delocalization and induction by CN groups. Figure 5 also illustrates that CN groups alone have a much higher negative charge concentration on the nitrogen (red color). In other cases, the GA value decreases from the initial to the final compound (i.e., fully substituted by TCNV groups) by around 35–30 kcal/mol. The more CN groups the initial acid contains, the lower the acidity increase is. This indicates that only a certain number of CN groups is necessary for a stable anion and strong acid. 

As a comparison, the GA values of hydrogen cyanide (HCN) **15H** and its altered analog [TCNV]H **16H** are included. HCN is a weak acid in the gas phase, having practically the same acidity as isocyanic acid [CNO]H **1H**. The most stable **T1** tautomer of [TCNV]H is 18.7 kcal/mol stronger acid than HCN. The most stable tautomers are both CH-acids; in the case of **16H**, the negative charge can delocalize into only one CN group.

The anionic acid B_12_(CN)_12_H^–^ **17H** is as strong as cyanoform. The B_12_(CN)_12_H_2_ **18H** acid is almost as strong as [B-TCNV_4_]H, **14H**. Carborane acid, **19H**, with 12 CN groups, is 4.4 kcal/mol stronger acid than **14H**. 

Tricyanovinyl alcohol [O-TCNV]H **2H** and cyanoform [(CN)_3_C]H **6H** are considered strong acids, and they have very similar acidities in the gas phase and the solution. Three CN groups seem to work as efficiently even though the protonation site (O vs. N) and the conformation are different. 

### 2.2. Solution Phase Acidities

Computational solution phase p*K*_a_ values in acetonitrile (MeCN) and dimethylsulfoxide (DMSO) were obtained using the COSMO-RS approach [37,38,39,40]. COSMO-RS is generally unsuitable to get absolute p*K*_a_ values; therefore, COSMO-RS p*K*_a_ values must be corrected using experimental p*K*_a_ values for similar acids [41]. For this reason, 12 auxiliary acids, whose p*K*_a_ values in MeCN are known, were used in the correlation to obtain corrected p*K*_a_(MeCN) values for the title compounds (Table 5 and Figure 6). Most of the known p*K*_a_(MeCN) values were, in fact, not directly measured in MeCN but were obtained from the correlation between p*K*_a_(DCE) and p*K*_a_(MeCN) values [3]. It means that the literature p*K*_a_(MeCN) values are already obtained from the correlation. All 12 auxiliary acids were 2-substituted-1,1,3,3-tetracyanopropenes (2-X-TCNP); **T4** tautomers of these acids were the most stable forms and only those tautomers were used to calculate COSMO-RS p*K*_a_ values. In the case of 3,4-(MeO)_2_-C_6_H_4_- and CN-CH_2_-substituted TCNP-s, two different conformers were used for COSMO-RS p*K*_a_ values. For the title compounds, different tautomers and conformers were used to calculate COSMO-RS p*K*_a_ values.

In addition to 12 auxiliary acids, 5 title compounds, whose literature p*K*_a_(MeCN) values are also known, were also included in the correlation to correct COSMO-RS values. As seen, two of those compounds, [O-TCNV]H **2H** and [(CN)_4_B]H **10H**, deviate significantly from the correlation line (Figure 6). **2H** is an OH acid, which does not suit the correlation of NH-acids, and hydrogen borates, even though protonated on the CN group, can deviate because of the peculiarity of the C-N-H fragment (see below).
p*K*_a_(MeCN) LITERATURE = 1.003 (0.052) · p*K*_a_(MeCN) COSMO-RS + 2.14 (0.11)*r*^2^ = 0.966; *S* = 0.416; n = 15(4)

Correlation 4 contains only CN acids, whose protonation site is in the CN group. It is not the most proper way to correct p*K*_a_(MeCN) values for OH (**2H**) and CH (**9H**) acids, and also, perhaps hydrogen borates (**10H**–**14H**) would need a different correlation; however, the scarcity of data does not allow creating an individual specific correlation equation for every group of compounds. Therefore, Equation (4) was used to obtain corrected computational p*K*_a_(MeCN) values for all the studied compounds. Most of the values are consistent with the literature data. It is assumed that even if the absolute corrected p*K*_a_(MeCN) values deviate from the actual p*K*_a_ values, then relative p*K*_a_ values for hydrogen borates, as well as cyclic carboranes and boranes, should be consistent with each other. It is also possible that the literature p*K*_a_(MeCN) value of **10H** is deviating since it is already in the literature obtained from the correlation.

p*K*_a_(DMSO) values in the current work are obtained from p*K*_a_(MeCN) values by averaging two different values calculated from previously published p*K*_a_(MeCN) *vs* p*K*_a_(DMSO) correlation equations [34,42]. COSMO-RS p*K*_a_(DMSO) values are also available; the correlation between acquired p*K*_a_(DMSO) values and COSMO-RS p*K*_a_(DMSO) values is very good (see SM), and it could be used to correct COSMO-RS p*K*_a_ values. 

p*K*_a_ values for 1,2-dichloroethane (DCE) were also obtained from the p*K*_a_(MeCN) values using a previously published correlation [3,36].

It is seen from Table 3 that our method strongly downgrades the acidity of HCN in MeCN (5 units) and DMSO (4 units) compared to the literature values. The literature p*K*_a_ value in MeCN is obtained computationally, and the p*K*_a_ value in DMSO is directly measured. The p*K*_a_ value of HCN depends very strongly on the solvation because its conjugate base is a very small anion. Computational methods that do not take solvation specifically into account, as the method used in our work, can give considerably wrong values. The p*K*_a_ values of HCN should be under deeper investigation in the future. 

### 2.3. Structural Details

In Figure 7, the structures of the initial anions and their conjugate acids are presented. Anion **1** is stick-like, **3** is bent, **6** is planar, and **10** is a tetrahedron. The most stable tautomer in the case of all initial acids is **T2**, i.e., the proton attached to the nitrogen on the CN group. H-N-C is bent except for the hydrogen borates, where the angle is almost always close to 180°. 

The geometry of other anions corresponds to their parent initial anion. TCNV groups are always planar, and the angles between substituents are close to the C-X-C angle shown in Figure 7. Angles between TCNV groups are generally larger than the angles between CN groups.

It is interesting to see how the most stable forms of anions that contain two TCNV groups differ from each other (Figure 8). The more single CN groups are bound directly to the central atom, the more the TCNV groups are compressed. *The anti-anti* conformer of anion **5** is 4.2 kcal/mol (BP86/def-TZVP results) more stable than the *syn-syn* conformer. *Anti-syn* **8** is 0.6 and 0.7 kcal/mol more stable than *syn-syn* and *anti-anti*, respectively. *Syn-syn* **12** is only 0.1 kcal/mol more stable than *anti-syn*, but *anti-anti* conformer already has 0.8 kcal/mol higher energy. This shows that the fewer substituents the anion contains, the more rigid the structure is. The most stable conformers of the corresponding conjugate acids of the anions **5** and **8** are the same. For **12H-T2**, the most stable conformer is *anti-syn*, and other conformers are 1.1 kcal/mol less stable. 

In Figure 9, the geometries of the bulkiest anions are shown. The C-C-C angles of the symmetrical anion **9** are all 119.9°, and the central part containing four carbon atoms is planar; only TCNV groups are bent out of the plane at 39.6°. Tetrahedral anion **14** has C-B-C angles all between 109.2 and 109.7°. The surface of the charge density shows that the negative charge concentration is highest on the N atoms of CN groups, but the charge is very effectively delocalized. 

Further analysis of the structures and energetics for each compound group is presented in the Appendix A.

## 3. Materials and Methods

QM calculations were carried out using TURBOMOLE 7.2 [43] and Gaussian 16 (T = 298.2 K, *p* = 1.0 atm) [44] The geometries of all compounds were created and optimized using the DFT method at the BP86/def-TZVP level. For all optimized geometries, vibrational spectra were computed to ensure that the optimized geometries correspond to the true energy minima. Imaginary frequencies in some could not be removed by re-optimization and were ignored. For the most stable compounds, other basis sets (see Table 4) were also used, geometries were optimized, and frequency calculations were carried out with the same level. A conformational search was carried out with COSMOconf 2021 (version 21.0) [45], and p*K*_a_ values were computed using COSMOthermX19 (version 19.0.4) [45] software. Computational methods were used in this work and are described in detail in Appendix A.

## 4. Conclusions

GA values for more than 300 different tautomers and conformers of polycyanated compounds were calculated using the DFT BP86/def-TZVP method. Final “selected GA values”, provided only for the most stable acid-base pairs, are calculated at the B3LYP/6-311 + G(d,p) level to be able to compare the GA values with previous works.

The most stable tautomers of initial CN-substituted compounds are tautomers protonated on the terminal CN group. These are isocyanic acid **1H**, iminomethylenecyanamide **3H**, 2-(iminomethylene)propanedinitrile **6H**, and hydrogen tetracyanoborate **10H**. The extensive substitution of CN groups by TCNV groups leads to the compounds having increased acidity for 30–50 kcal/mol in the gas phase, reaching nearly 230 kcal/mol. The acidity increase in MeCN is 9 to 19 p*K*_a_ units reaching −13. The most stable tautomers are now the compounds protonated on the central atom (O, N, C) or the CN group closest to the central atom (B). [B-TCNV_4_]H is almost as strong as fully cyanated carborane acid, being approximately only a 5 kcal/mol weaker acid. The strongest acids in the current work can be considered super- and hyperacids.

## Figures and Tables

**Figure 1 molecules-28-08157-f001:**
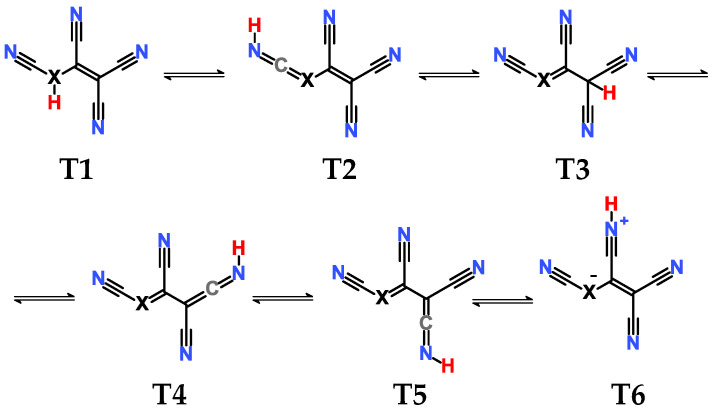
Schematic presentation of tautomers calculated in the current work. **T** denotes a tautomer which is referred to everywhere in the text. In case X = B, the tautomers are zwitterions rather than neutral compounds.

**Figure 2 molecules-28-08157-f002:**
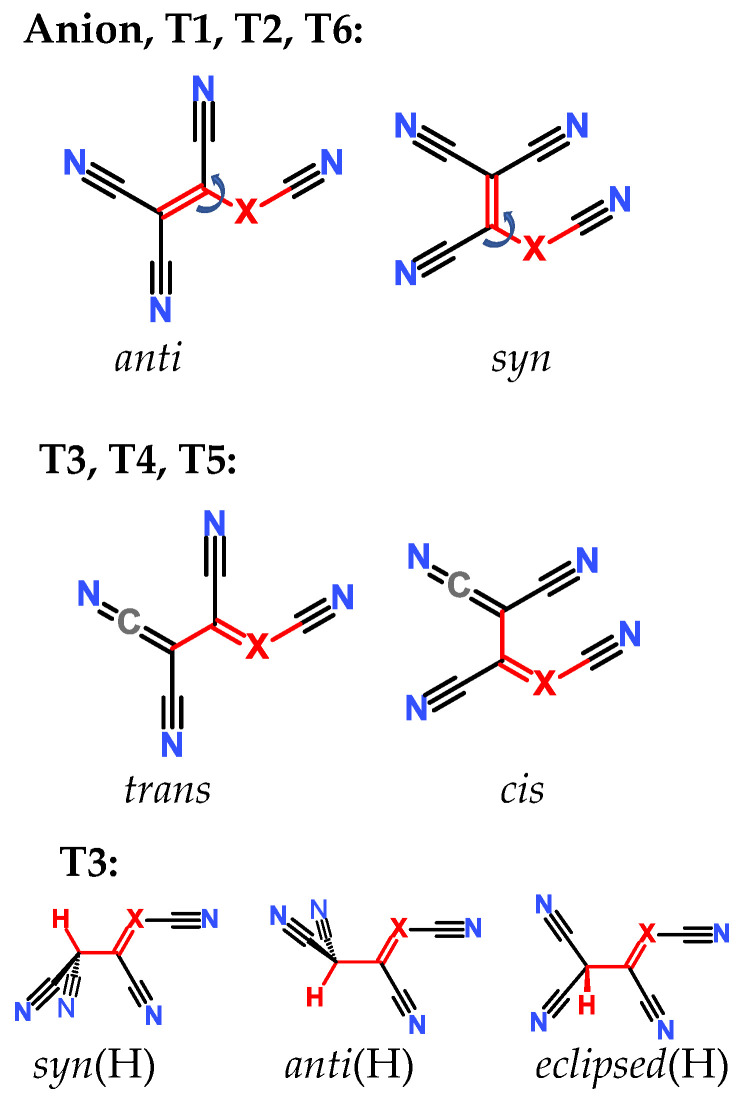
Different diastereomers calculated in the current work. **T1**–**T6** refers to tautomers.

**Figure 5 molecules-28-08157-f005:**
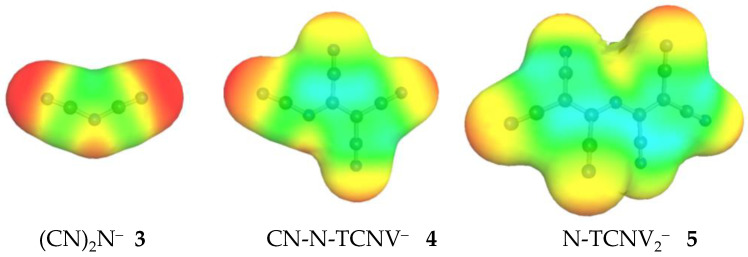
The surface of the charge density of anions **3**, **4,** and **5**, computed using the COSMO-RS approach. The red area indicates a negative partial charge on the CN groups and the green area indicates close to zero partial charge.

**Figure 6 molecules-28-08157-f006:**
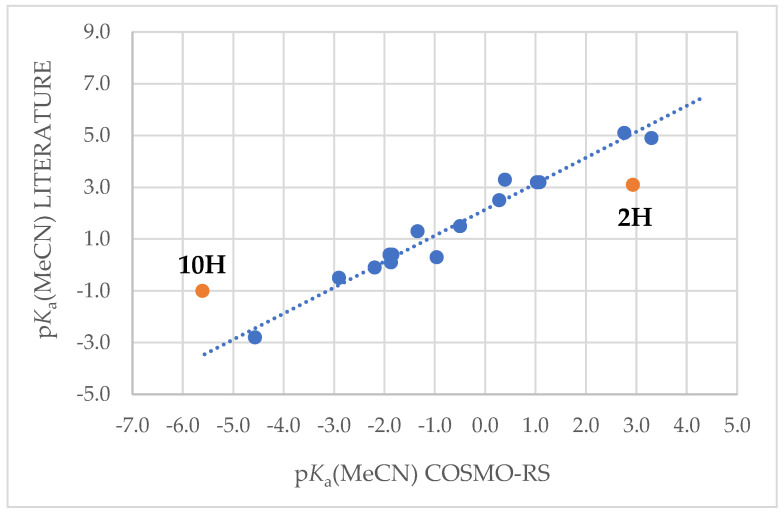
Blue dots are used to create the correlation between literature and COSMO-RS computational values (Equation (4)) of the compounds listed in Table 5, whereas red points (**2H** and **10H**) have been omitted from the correlation.

**Figure 7 molecules-28-08157-f007:**
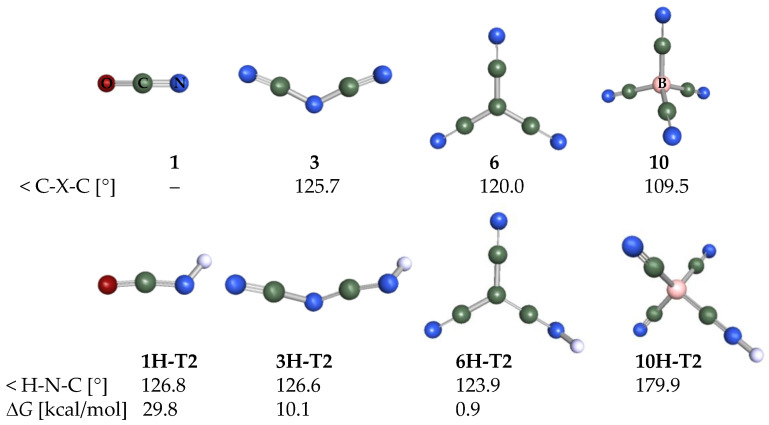
Geometries of initial anions **1**, **3**, **6**, and **10**, and their most stable acid forms. H-N-C angles are included. Δ*G* values (kcal/mol) show how much the **T2** tautomer is more stable relative to the **T1** tautomer at BP86/def-TZVP level. Here and in other figures (also in Appendix A) red color corresponds to oxygen, green to carbon, blue to nitrogen and pink to boron atoms.

**Figure 8 molecules-28-08157-f008:**
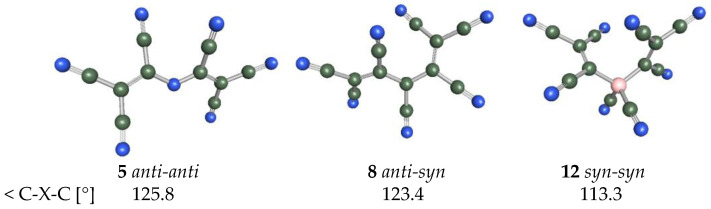
The most stable conformers of anions **5**, **8**, and **12** containing two TCNV groups.

**Figure 9 molecules-28-08157-f009:**
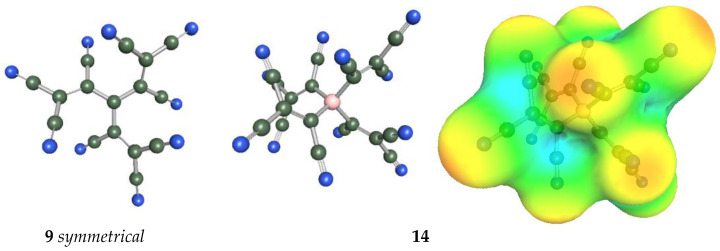
Geometry of anions **9** and **14**. The surface of the charge density of anion **14**.

**Table 1 molecules-28-08157-t001:** Schematic presentation of the anions of title compounds studied in the current work. In the text, anions’ conjugate acids are denoted as NumberH with tautomer number (**T1**–**T6**), if necessary.

Initial Compound	Step 1	Step 2	Step 3	Step 4
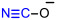	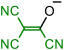			
**1**	**2**			
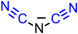	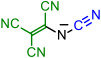	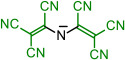		
**3**	**4**	**5**		
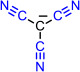	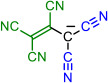	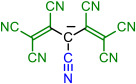	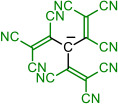	
**6**	**7**	**8**	**9**	
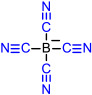	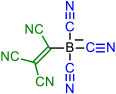	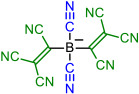	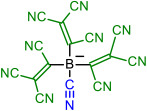	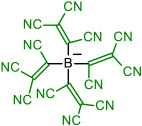
**10**	**11**	**12**	**13**	**14**

**Table 2 molecules-28-08157-t002:** Other compounds as anions which are used for comparison.

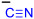	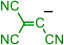	[B_12_(CN)_12_]^2–^	[B_12_(CN)_12_]H^–^	[CB_12_(CN)_12_]^–^
**15**	**16**	**17**	**18**	**19**

**Table 5 molecules-28-08157-t005:** A list of auxiliary acids used to obtain corrected p*K*_a_(MeCN) values, **2H** and **10H**, is excluded from the correlation. X-TCNP = 2-X-1,1,3,3-tetracyanopropene [3].

Compound		LITERATURE p*K*_a_(MeCN)	COSMO-RS p*K*_a_(MeCN)
NH_2_-TCNP		3.30	4.9
3,4-(MeO)_2_-C_6_H_3_-TCNP		1.08	3.2
4-MeO-C_6_H_4_-TCNP		1.03	3.2
Ph-TCNP		0.28	2.5
3-CF_3_-C_6_H_4_-TCNP		−0.50	1.5
H-TCNP		−1.34	1.3
Br-TCNP		−1.90	0.4
3,5-(CF_3_)_2_-C_6_H_3_-TCNP		−1.84	0.4
Cl-TCNP		−1.87	0.1
CN-CH_2_-TCNP		−2.19	−0.1
CF_3_-TCNP		−2.90	−0.5
Me-TCNP		0.39	3.3
[N-TCNV_2_]H	**5H**	−0.96	0.3
[(CN)_3_C]H	**6H**	2.76	5.1
[(CN)_2_C-TCNV]H	**7H**	−4.57	−2.8
[O-TCNV]H	**2H**	3.1	2.93
[(CN)_4_B]H	**10H**	−1.0	−5.61

## Data Availability

Publicly available datasets were analyzed in this study. This data can be found here: https://doi.org/10.23673/re-436.

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
