# Peer review of "Step-by-Step Replacement of Cyano Groups by Tricyanovinyls—The Influence on the Acidity"

_molecules, 2023, doi:10.3390/molecules28248157_

Round 1

Reviewer 1 Report

Comments and Suggestions for Authors

Review is attached.

Comments on the Quality of English Language

See comments in the manuscript review. 

Author Response

I reorganized the manuscript slightly.  The section "Structural details," as it was, was placed in Supporting Material. Only a short section remained in the main text, which is still at the end of the article. I wanted to present the GA and pKa values first rather than structures. The stability of tautomers is available from the beginning of the manuscript, which is enough to understand the text.

G4(MP2) GA values are probably the best, but their calculation is complicated for larger compounds; in fact, as I discovered, I do not have G4(MP2) GA values for all the title compounds. Besides, I wanted to use B3LYP/6-311+G(d,p) values, even though they slightly underestimate the acidities compared to G4(MP2) values, because they are directly comparable with most of the work by Koppel and Maksic and Vianello.

Definitions of GA and PA, as well as pKa values, are now placed in the main text. The terms “Acidity increase” and “GA value increase” were solved.

The term “Strong acidity” is strange to use. The whole sentence was rewritten; perhaps it is correct now.

The data from the NIST database was added to Table 3.

“Sigma surface” was redefined.

Figure 5 was simplified, but it remained in the main text.

“Selected values” was used instead of “assigned values.”

Gaussian T and p were stated.

Unusual wording was rewritten.

The dilemma concerning isocyanic acid vs cyanic acid was most probably solved.

The conclusion was slightly rewritten.

Reviewer 2 Report

Comments and Suggestions for Authors

In this manuscript, the effects of cyano (CN) and 1,2,2-tricyanovinyl 9 (TCNV) groups on the acidity of some compounds are investigated, computationally. The acidities of the compounds are compared in the gas phase and solutions. The level of theory and computational methods are reasonable and so the reported data are reliable. However, the paper is hard to follow as the nomenclature of the compounds is confusing.

-The author has used numbers 1, 2, 3, … for the conjugated bases (the anionic compounds) and 1H, 2H, 3H, … for the neutral corresponding acids. I would use 1, 2, 3 for the acids and [1-H]-, [2-H]-, and [3-H]- for the conjugated bases.

- Proton affinity (PA) is an index of basicity in the gas phase, however, we can use it as an index of acidity when it is calculated for the protonation of a conjugated base. In Table 3, PA is reported for neutral acids 1H, 2H, 3H, …. that I think is not correct.  The author can simply use ΔHacid for these neutral acids instead of PA.

- In most of the Tables, the computational method is not clear, although it has been mentioned in SM, but it is still confusing. The computational method can be added to the Table caption.

- I recommend that the author compare the gas phase acidity of the studied compounds with the acidity of reported polycyano compounds (10.1002/ejoc.200500153) and acidity of compounds containing B atom (10.1016/j.cplett.2020.138207).

The manuscript can be considered for publication after revision.

Author Response

-The author has used numbers 1, 2, 3, … for the conjugated bases (the anionic compounds) and 1H, 2H, 3H, … for the neutral corresponding acids. I would use 1, 2, 3 for the acids and [1-H]-, [2-H]-, and [3-H]- for the conjugated bases.

Response: I thank the referee for the idea of using better numeration to make it easier to read; however, the suggested way to use the numeration may confuse the reader since 1-H may refer to 1 with the attached H atom. Therefore, I did stick with the current numbering. I also tried to clarify the text by adding the compound formulas.

- Proton affinity (PA) is an index of basicity in the gas phase, however, we can use it as an index of acidity when it is calculated for the protonation of a conjugated base. In Table 3, PA is reported for neutral acids 1H, 2H, 3H, …. that I think is not correct.  The author can simply use ΔHacid for these neutral acids instead of PA.

Response: It is now corrected and moved to the main text as an introduction. I did not want to use DHacid and DGacid to distinguish between DG, which I also used.

- In most of the Tables, the computational method is not clear, although it has been mentioned in SM, but it is still confusing. The computational method can be added to the Table caption.

Response: The computational method was now added to all the tables.

- I recommend that the author compare the gas phase acidity of the studied compounds with the acidity of reported polycyano compounds (10.1002/ejoc.200500153) and acidity of compounds containing B atom (10.1016/j.cplett.2020.138207).

Response: These two references were added to the manuscript.

Reviewer 3 Report

Comments and Suggestions for Authors

In the present manuscript, the author computationally addressed the acidity of several molecular species in which the cyano group is replaced by the tricyanovinyl group. It was found that the tricyanovinyl group has a greater influence on acidity than the cyano group, indicating that stronger acids can be achieved by utilizing the tricyanovinyl group instead of the cyano group. The study meticulously examined all the structural details regarding tautomerism and conformational space. Furthermore, the influence of different computational models on gas acidity was explored.

In my opinion, the manuscript is well written and could be published after minor, mostly technical, revisions:

1.      Table 3 is too wide and partially empty. Perhaps the scarce experimental numbers could be placed in the same column where computational numbers are presented. However, experimental values could be distinguished by enclosing them in round, square, or curly brackets.

2.      Figure 5 is challenging to follow. While I understand and appreciate the effort to include as much data as possible in one place, the figure's legend and explanation should be made more clear and understandable. I spent at least 20 minutes trying to understand which number corresponds to which method(structure). Fortunately, some of these numbers are presented in Table 5. However, since Table 5 comes after Figure 5, readers will spend too much time trying to decipher what Figure 5 represents. In conclusion, Figure 5 can stay as it is, but a more detailed legend and textual explanation are needed.

3.      I am somewhat concerned that the GA values obtained by DFT methods are sometimes significantly different from those obtained by G4(MP2). Since G4(MP2) should be much more accurate, the source of the problem may lie in the DFT results. Are the author sure that there is no biradical character in the anionic species? Standard DFT models do not properly address biradical character, which could be the source of error. Have the author tested the stability of the wavefunction for anionic species?

4.      There are some typos here and there (e.g., 'condormation' -> 'conformation'; line 207), so a spellchecker should be applied.

Comments on the Quality of English Language

The English language in the text is accurate and well-written. There are some typos which could be corrected using spellchecker.

Author Response

  1. Table 3 is too wide and partially empty. Perhaps the scarce experimental numbers could be placed in the same column where computational numbers are presented. However, experimental values could be distinguished by enclosing them in round, square, or curly brackets.

Response: The table has now reformatted the way the referee suggests.

  1. Figure 5 is challenging to follow. While I understand and appreciate the effort to include as much data as possible in one place, the figure's legend and explanation should be made more clear and understandable. I spent at least 20 minutes trying to understand which number corresponds to which method(structure). Fortunately, some of these numbers are presented in Table 5. However, since Table 5 comes after Figure 5, readers will spend too much time trying to decipher what Figure 5 represents. In conclusion, Figure 5 can stay as it is, but a more detailed legend and textual explanation are needed.

Response: I find Figure 5 to contain useful information; therefore, I would still like to keep Figure 5 in the main text. Instead of adding more labels, I simplified it and wrote slightly more information about it.

  1. I am somewhat concerned that the GA values obtained by DFT methods are sometimes significantly different from those obtained by G4(MP2). Since G4(MP2) should be much more accurate, the source of the problem may lie in the DFT results. Are the author sure that there is no biradical character in the anionic species? Standard DFT models do not properly address biradical character, which could be the source of error. Have the author tested the stability of the wavefunction for anionic species?

Response: I calculated the GA values based on G4(MP2) method using the wrong energies from Gaussian output. The G4(MP2) (as well as W1RO and CBS-APNO) GA values are now corrected and match quite well with other methods. Tables in SM and data in the data repository were corrected as well.

  1. There are some typos here and there (e.g., 'condormation' -> 'conformation'; line 207), so a spellchecker should be applied.

Response: This and other typos were corrected.

Round 2

Reviewer 1 Report

Comments and Suggestions for Authors

Reviewer report

Manuscript ID molecules-2711695Rev 2

Title: Step-by-step replacement of cyano groups by tricyanovinyls – the influence on the acidity

Author: Agnes Kütt

Section: Computational and Theoretical Chemistry

Special Issue: Computational and Theoretical Studies on Isomeric Organic Compounds

Several corrections were done along my previous comments, but I am disappointed that a detailed list of changes was not provided. The manuscript is now easier to follow. Some improvement of the terminology problem of the gas-phase acidity (e.g. decrease of PA or GA values meaning higher acidity) was done, but there are again some wrong assertions, see below. Some responses to my comments are rather surprising, for example “in fact, as I discovered (sic), I do not have G4(MP2) GA values for all the title compounds.” There are still some small English imperfections, which I do not wish to correct in detail.

Some corrections must be checked, as described below.

Lines 24-26: “PA of an anionic base is defined as the negative of the enthalpy change (ΔHacid) the gas-phase reaction (1) between a proton and the anion to produce the neutral conjugate acid of the anion.” The inverse of reaction (1) is described, so there is a contradiction. ΔHacid is simply the enthalpy of reaction (1).

Lines 26-27: “GA is defined as the negative of the Gibbs free energy (ΔGacid) for 26 reaction 1”. As for the enthalpy, ΔGacid is directly the Gibbs energy of reaction (1), not the negative value.

Lines 27-28: “The GA includes the entropic term, is directly measurable and comparable with experimental values,…”. If the entropic term is measurable (in fact not so easily), it is already experimental.

Equation (3), the activities of the species are not defined.

Lines 176-178, “The selected B3LYP/6-311+G(d,p) GA values seem to give the lowest acidities; however, they match pretty well with 177 G4(MP2) GA values.” The ambiguity (or error) still persists between “lowest acidities” and the data. What is seen in Table 3, is that all B3LYP values (except for 3H) are lower that G4(MP2), and also lower than most “Turbomole” values, indicating in fact “higher acidities”.

The Conclusion (four lines) is still not up to the work described in the manuscript.

“Supporting” and “Supplementary” (materials) are used interchangeably.

Comments on the Quality of English Language

There are minor corrections.

Author Response

Several corrections were done along my previous comments, but I am disappointed that a detailed list of changes was not provided. The manuscript is now easier to follow. Some improvement of the terminology problem of the gas-phase acidity (e.g. decrease of PA or GA values meaning higher acidity) was done, but there are again some wrong assertions, see below. Some responses to my comments are rather surprising, for example “in fact, as I discovered (sic), I do not have G4(MP2) GA values for all the title compounds.” There are still some small English imperfections, which I do not wish to correct in detail.

Response: What does the referee mean by “detailed list”? The reason the detailed list was not provided was that the referee sent an encrypted (?) PDF files that I could not copy-paste. The numbering of the list would have helped a little. It was very annoying, and as much as I tried, I answered most of the bullets on the list. With the aid of the editor I was now capable to get the review as text, and now I can give the detailed list!

Some corrections must be checked, as described below.

Lines 24-26: “PA of an anionic base is defined as the negative of the enthalpy change (ΔHacid) the gas-phase reaction (1) between a proton and the anion to produce the neutral conjugate acid of the anion.” The inverse of reaction (1) is described, so there is a contradiction. ΔHacid is simply the enthalpy of reaction (1).

Lines 26-27: “GA is defined as the negative of the Gibbs free energy (ΔGacid) for 26 reaction 1”. As for the enthalpy, ΔGacid is directly the Gibbs energy of reaction (1), not the negative value.

Lines 27-28: “The GA includes the entropic term, is directly measurable and comparable with experimental values,…”. If the entropic term is measurable (in fact not so easily), it is already experimental.

Response: The terminology concerning GA and PA was rewritten.

Equation (3), the activities of the species are not defined.

Response: Activities are now defined.

Lines 176-178, “The selected B3LYP/6-311+G(d,p) GA values seem to give the lowest acidities; however, they match pretty well with 177 G4(MP2) GA values.” The ambiguity (or error) still persists between “lowest acidities” and the data. What is seen in Table 3, is that all B3LYP values (except for 3H) are lower that G4(MP2), and also lower than most “Turbomole” values, indicating in fact “higher acidities”.

Response: The part of the text has now been rewritten.

The Conclusion (four lines) is still not up to the work described in the manuscript.

Response: The conclusion was supplemented.

“Supporting” and “Supplementary” (materials) are used interchangeably.

Response: “Supplementary Material” was now used everywhere.